# Phosphorylation induces sequence-specific conformational switches in the RNA polymerase II C-terminal domain

Eric B. Gibbs[1], Feiyue Lu[2,3], Bede Portz[3], Michael J. Fisher[3], Brenda P. Medellin[4,5], Tatiana N. Laremore[2], Yan Jessie Zhang[4,5], David S. Gilmour[3] & Scott A. Showalter[1,3]

The carboxy-terminal domain (CTD) of the RNA polymerase II (Pol II) large subunit cycles through phosphorylation states that correlate with progression through the transcription cycle and regulate nascent mRNA processing. Structural analyses of yeast and mammalian CTD are hampered by their repetitive sequences. Here we identify a region of the *Drosophila melanogaster* CTD that is essential for Pol II function *in vivo* and capitalize on natural sequence variations within it to facilitate structural analysis. Mass spectrometry and NMR spectroscopy reveal that hyper-Ser5 phosphorylation transforms the local structure of this region via proline isomerization. The sequence context of this switch tunes the activity of the phosphatase Ssu72, leading to the preferential de-phosphorylation of specific heptads. Together, context-dependent conformational switches and biased dephosphorylation suggest a mechanism for the selective recruitment of *cis*-proline-specific regulatory factors and region-specific modulation of the CTD code that may augment gene regulation in developmentally complex organisms.

[1] Department of Chemistry, The Pennsylvania State University, University Park, Pennsylvania 16802, USA. [2] Huck Institutes of the Life Sciences, The Pennsylvania State University, University Park, Pennsylvania 16802, USA. [3] Center for Eukaryotic Gene Regulation, Department of Biochemistry and Molecular Biology, The Pennsylvania State University, University Park, Pennsylvania 16802, USA. [4] Department of Molecular Biosciences, University of Texas at Austin, Austin, Texas 78712, USA. [5] Institute for Cellular and Molecular Biology, University of Texas at Austin, Austin, Texas 78712, USA. Correspondence and requests for materials should be addressed to S.A.S. (email: sas76@psu.edu).

The carboxy-terminal domain (CTD) of Rpb1, the largest subunit in RNA polymerase II, is an essential regulator of eukaryotic gene expression. This intrinsically disordered protein (IDP), consisting of multiple tandem repeats of the consensus sequence $(Y^1S^2P^3T^4S^5P^6S^7)$, acts as a scaffold for the recruitment of factors required for transcription, mRNA biogenesis and modification of the chromatin structure[1]. Tight control over the spatial and temporal recruitment of CTD-associated factors is regulated at the molecular level in part by CTD-specific kinases and phosphatases, which generate dynamic patterns of post-translational modifications (PTMs) collectively referred to as the 'CTD code'[2]. While much is known about how heptads matching the consensus sequence contribute to gene expression[1], heptads that deviate from the consensus are found in all eukaryotes whose sequences are known. The number and complexity of these non-consensus heptads roughly correlate with developmental complexity[3]. Expression of genes involved in multicellularity were affected by mutating non-consensus heptads of mouse cells in culture[4], despite data indicating that the non-consensus heptads are not essential for the viability of human cells grown in culture[5]. Likewise, deletion of a small region encompassing several non-consensus heptads caused severe developmental defects and growth retardation in mice[6], demonstrating that non-consensus heptads may contribute to development and cellular differentiation.

Despite the wealth of information detailing CTD function, little is known about the molecular basis of the CTD code. The established view is that the intrinsically disordered nature of the CTD renders its structure and interactions non-specific, to be dictated predominantly by its PTM status[7]. Prior structural studies, focussing on short CTD peptides composed entirely of consensus heptad repeats, revealed turn and coil structures that have been extrapolated to represent the entirety of the CTD[7,8]. The repetitive amino-acid sequence comprising the CTD has been a major obstacle in studying its structure because it prevents heptad-specific interpretation of both mass spectra and NMR spectra. Therefore, recent mass spectrometry-based investigations have resorted to introducing mutations that facilitate analysis of specific regions of the CTD[9,10]. However, it is unclear how best to interpret these results molecularly because the impacts of mutations on IDP structure are often difficult to predict. Here we turn to *Drosophila*, which is unique among commonly used model organisms in that only 2 of its 42 heptads precisely match the consensus sequence. This feature has allowed us to use NMR and mass spectrometry to precisely monitor PTM patterns and local structural features in the context of a natural CTD sequence.

## Results

**Transgenic flies reveal a CTD region needed for development**. To identify a functionally significant region of the CTD for our structural analysis, we tested the ability of ectopically expressed derivatives of Rpb1 to rescue the lethality caused by inhibiting the expression of endogenous Rpb1. When a transgenic fly line expressing RNA-mediated interference (RNAi) against endogenous Rpb1 in response to the GAL4 activator is mated to a fly line that ubiquitously expresses GAL4, no adult progeny are produced. This lethality is overcome by co-expressing a wild-type version of Rpb1 that has been rendered resistant to the RNAi through synonymous substitutions in the Rpb1-coding sequence (Fig. 1a). To identify regions of the CTD essential for the development of an adult fly, several deletions were made in the CTD of the RNAi-resistant form of Rpb1 and tested for their ability to rescue the lethality associated with depleting the endogenous Rpb1 (Fig. 1b, Supplementary Fig. 1b). Western blot

analysis with antibody against the ectopically expressed Rpb1 indicated that each derivative was expressed at comparable levels in tissues derived from pupae, indicating that each Rpb1 variant was equally stable (Fig. 1c, Supplementary Fig. 1c). The CTDΔ2 mutant was the only one that failed to produce adult flies (Fig. 1d, Supplementary Table 1). Strikingly, this region of the *Drosophila* CTD is the most highly conserved among higher eukaryotes (Fig. 1e, Supplementary Fig. 1b), suggesting functional necessity.

**Hyper-phosphorylation of recombinant CTD2′ by P-TEFb.** Having identified a region of the *Drosophila* CTD that is essential for normal development, we next sought to incorporate the region removed in the CTDΔ2 mutant into a recombinant construct displaying the properties necessary for high-resolution structural biology. We selected a CTD construct, CTD2′-containing residues 1,657–1,739 of the Rpb1 polypeptide sequence (Fig. 2a). In order to explore how phosphorylation impacts the structure of the CTD, we used *D. melanogaster* positive transcription elongation factor b (Dm P-TEFb) to hyper-phosphorylate CTD2′ *in vitro*[11]. Analysis of hyper-phosphorylated CTD2′ by mass spectrometry (MS) revealed successful incorporation of up to 12 phosphates per polypeptide (Fig. 2b). Phospho-site identification by tandem MS (MS/MS) led to the conclusion that our *in vitro* phosphorylation protocol predominantly generates high levels of Ser5 phosphorylation (Fig. 2a,c). Preferential phosphorylation *in vitro* of Ser5 over other amino acids in the CTD by human P-TEFb has been previously observed[12,13].

NMR spectroscopy was employed to augment and cross-check our MS-based phospho-site assignment, leading to the pattern depicted in Fig. 2a. NMR is well suited to this task due to its high sensitivity to the local chemical environments of individual residues, but backbone resonance assignment of disordered and repeat-containing polypeptides such as the CTD is often complicated by extreme signal overlap[14–16] (Fig. 3a). To circumvent this limitation, we and others have developed $^{13}C$ Direct-Detect NMR spectroscopy[17], which vastly improves resonance dispersion for intrinsically disordered regions (Fig. 3b) and also provides direct measurement of proline resonances (Fig. 3d)[18,19]. This allowed unambiguous backbone resonance assignment of CTD2′, including assignment of the proline residues, which comprise 23% of the polypeptide sequence. Assignments were mapped onto $^1H,^{15}N$ correlation spectra through standard triple resonance experiments. Certain amide proton and side-chain carbon resonances (Figs 2d,f and 3c) experienced downfield chemical shifts that were consistent with phosphorylation[20,21]. Correlated with these shifts, several proline residues adjacent to phosphorylation sites showed carbon chemical shift changes characteristic of *trans*-to-*cis* isomerization of the peptide plane (Figs 2d,f and 3d). In total, 10 *bona fide* phospho-sites were identified. Real-time NMR (RT-NMR) permitted the kinetic measurement of CTD2′ phosphorylation, revealing that for the seven internal heptads containing Ser5-Pro6 pairs phosphorylation proceeded at similar apparent rates and reached comparable levels upon saturation (90%) (Fig. 2e, Supplementary Fig. 2, Supplementary Table 2), with incomplete phosphorylation of the C-terminal repeat, and two internal repeats lacking a Ser5-Pro6 pair (Ser5 in YSPSSN and Thr5 in YTPVTPS). The observed rates are consistent with a distributive mechanism similar to that observed for the human P-TEFb[12]. Thus the *in vitro* phosphorylation reactions produce a nearly complete hyper-pSer5 state, in agreement with our MS data (Fig. 2a–c,f).

**Hyper-phosphorylation does not alter the scale of CTD2′.** We next turned to a detailed investigation of the effects hyper-phosphorylation has on the structure of the CTD2′ region. In order to test how compact CTD2′ is in solution, which may

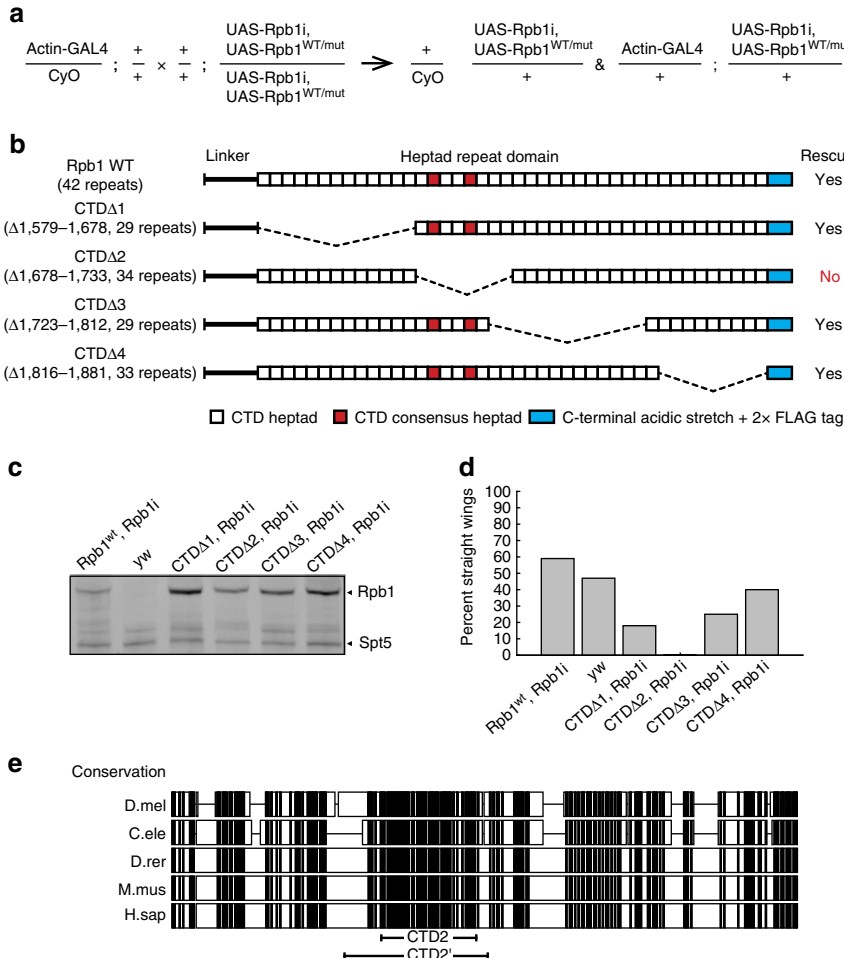

**Figure 1 | A highly conserved region in the *Drosophila* CTD is essential for viability and is targeted for Ser5 phosphorylation by Dm P-TEFb *in vitro*.**
(**a**) Schematic of the rescue assay: Ubiquitous expression of shRNA targeting Rpb1 (UAS-Rpb1i) by the Actin-GAL4 transgene is lethal and results in no straight-winged progeny. Rescue as a result of co-expression of an Rpb1 derivative (UAS-Rpb1^WT/mut) is indicated by the presence of straight-winged adults among the progeny. (**b**) Four internal deletion mutants of Rpb1 were expressed under Actin-GAL4 control to test for rescue of lethality caused by ubiquitous depletion of endogenous Rpb1 (*Actin-GAL4/ +; UAS-Rpb1i/ +*). Only CTDΔ2 failed to rescue. (**c**) Western blot analysis of the expression of Rpb1 derivatives: Tissues were collected from late pupae derived from the same genetic cross as described in **a**. Late pupae were analysed because these were produced by each of the matings, including the one with CTDΔ2. Ectopic Rpb1 expression was detected with FLAG antibody. Detection of Spt5 served as a loading control. (**d**) Percentages of straight-winged adults from each cross. For numbers of flies examined, see Supplementary Fig. 1c.
(**e**) Evolutionary conservation of the CTD across different species. Identical amino acids are denoted by black bars. The region encompassed by the CTDΔ2 deletion (CTD2) and recombinant protein CTD2′ (black lines, bottom) contain a highly conserved region.

impact its accessibility to CTD-binding factors, we collected small-angle X-ray scattering (SAXS) data on the unphosphorylated and hyper-pSer5 states (Fig. 4, Supplementary Table 3). In the unphosphorylated state, CTD2′ displayed an average $R_g$ of $28.0 \pm 0.7$ Å, while hyper-pSer5 CTD2′ displayed a similar average $R_g$ of $28.3 \pm 0.3$ Å using the Guinier approximation (Supplementary Fig. 3). For comparison, the ubiquitous CTD interaction domain ($\sim 140$ amino acids[22]) has an $R_g \sim 17$ Å; in contrast, the nucleosome core particle ($\sim 800$ amino acids and 146 DNA base-pairs) has an $R_g \sim 41$ Å, which demonstrates that CTD2′ is relatively expanded in solution. Similarly, pair-wise distance distributions revealed no significant increase in the maximum dimension ($D_{max}$) upon phosphorylation (Fig. 4b). These results are in good agreement with the dimensions predicted for an excluded volume random coil with the same number of monomers as CTD2′ (ref. 23). Independently, [31]P NMR spectroscopy revealed that the phosphates in the CTD were in the $-2$ charge state under our experimental conditions (Supplementary Fig. 4), and yet charge–charge repulsions did not

appear to impact the dimensions of the CTD. Incorporation of this data into a model for random-coil structure demonstrates that the median pSer5–pSer5 distance (approximately 18 Å) is likely to be greater than the Debye screening length under our experimental conditions (Supplementary Fig. 5), which accounts for the lack of chain expansion upon hyper-phosphorylation. In summary, our SAXS data demonstrate that the region of the CTD encompassed by CTD2′ experiences no significant phosphorylation-induced change in structure on the nanometer length scale. This suggests that the functional purpose of hyper-phosphorylation is not to effect a dramatic expansion of the CTD in order to modulate access by interacting factors but rather to change the pattern of binding motifs displayed on an already expanded structure.

**pSer5 induces sequence-dependent CTD proline isomerization.**
On the local scale of several to tens of amino acids, which is the size of the motifs most CTD-interacting domains recognize,

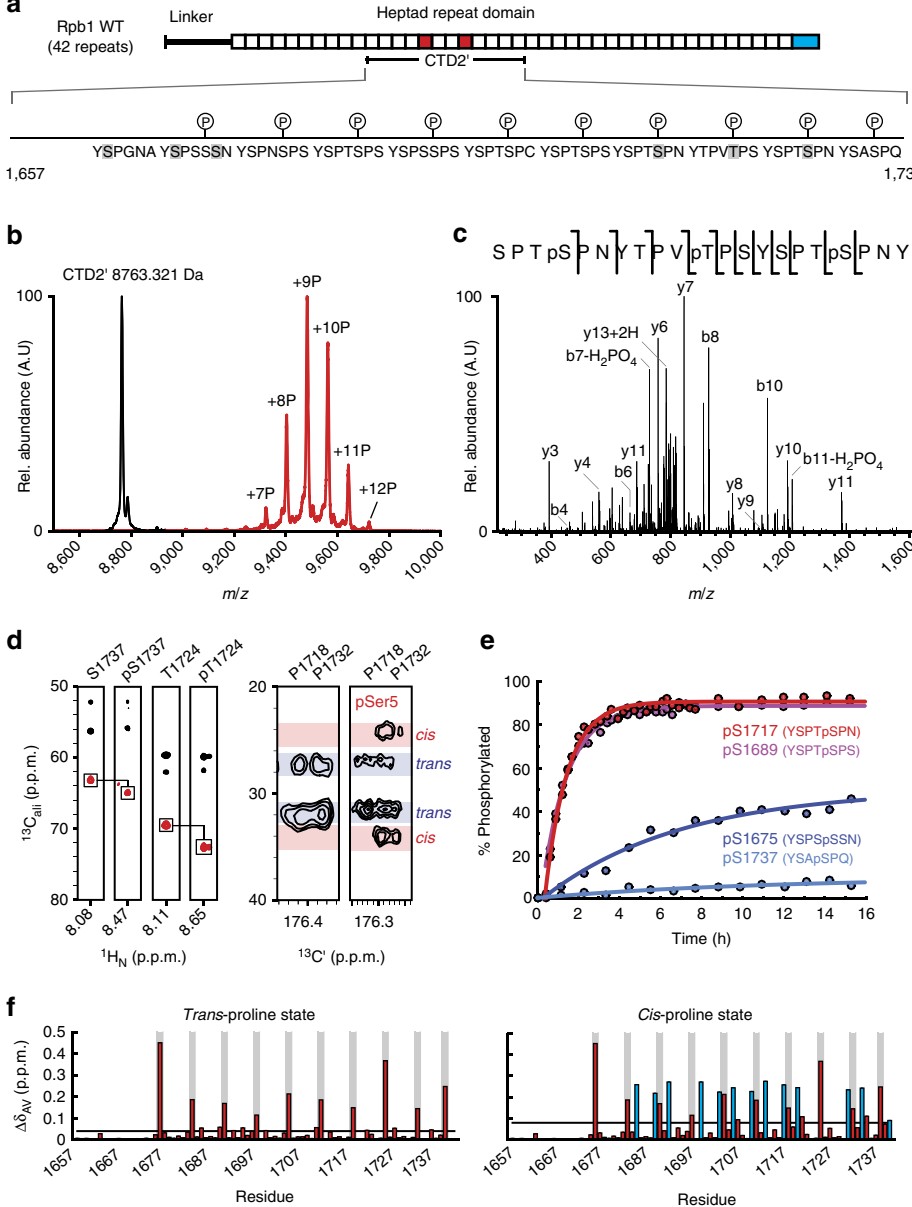

**Figure 2 | CTD2′ Ser5 phosphorylation probed by MS and NMR spectroscopy.** (**a**) Amino-acid sequence of CTD2′ displaying 99% confidence phospho-site assignments by MS/MS (grey boxes) and by NMR (P-symbols). MS/MS peptide coverage was complete. (**b**) Linear positive MALDI TOF MS of unphosphorylated CTD2′ (black) and hyper-pSer5 CTD2′ (red). (**c**) Representative spectrum from Nano-LC MS/MS analysis of hyper-pSer5 CTD2′. (**d**) Representative strips from 3D HNCACB spectra of unphosphorylated and hyper-pSer5 CTD2′ showing perturbation upon phosphorylation (left) and strips from 3D CCCON spectra of unphosphorylated and hyper-pSer5 CTD2′ showing pSer5-induced *trans*-to-*cis* isomerization of Pro6 (P1718 and P1732; right). (**e**) Representative kinetic traces for CTD2′ phosphorylation monitored by RT-NMR. (**f**) Average chemical shift perturbations for CTD2′ upon phosphorylation for the *trans*-proline-enriched (red) and *cis*-proline-enriched states (blue). Grey bars indicate pSer/pThr5 residues and the black line denotes the average perturbation.

phosphorylation has been shown to strongly perturb backbone dihedral angles[24–26], suggesting that phosphorylation could induce local structural perturbations in the CTD. Therefore, we used NMR to probe the backbone conformation of CTD2′. Secondary structure populations for the unphosphorylated state of CTD2′ were calculated from chemical shifts using the secondary structure calculation program δ2D (Supplementary Fig. 6a), revealing small populations of extended β/PPII character and a strong propensity for random coil-like conformations, consistent with our SAXS results and previous solution studies of CTD peptides[15]. Further, proline $C_\beta$ and $C_\gamma$ chemical shifts demonstrated a strong preference for the *trans*-proline confor-

mation (~95% *trans*-proline isomer) (Fig. 5a, Supplementary Fig. 6b). Thus our NMR and SAXS data are strongly consistent with Dm CTD free in solution adopting a spatially heterogeneous ensemble that is highly dynamic on the approximately nanosecond timescale. This observation is consistent with prior reports which concluded that short CTD-derived peptides are predominantly random in solution, lacking long-range order or temporally persistent tertiary structure[14,15,27,28]. In summary, the unphosphorylated state CTD2′ is highly disordered, temporally dynamic and contains nearly all *trans*-proline.

Surprisingly, in the hyper-pSer5 state, CTD2′ displayed a twofold enrichment of *cis*-proline content averaged over all

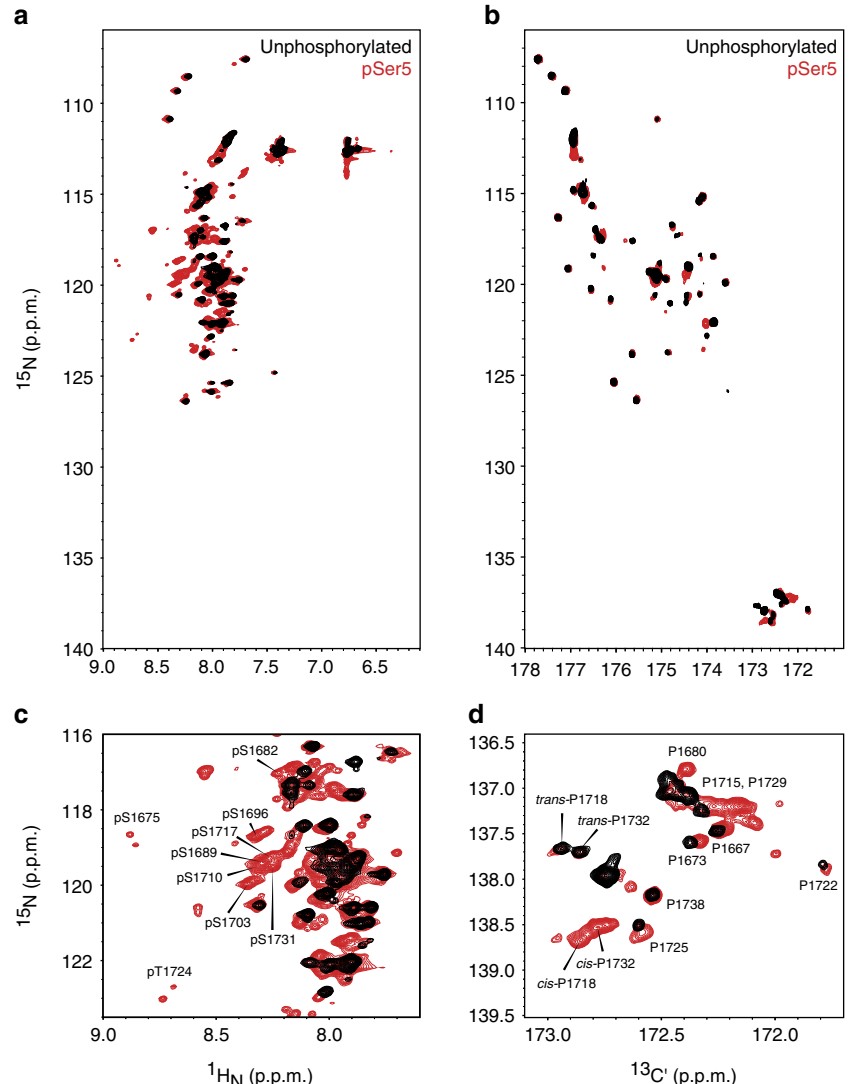

**Figure 3 | Phospho-sites and proline isomerization in hyper-pSer5 CTD2′ probed by NMR spectroscopy. (a)** 2D $^1$H-$^{15}$N correlation spectra of unphosphorylated (black) and hyper-pSer5 (red) CTD2′. **(b)** 2D $^{13}$C′-$^{15}$N correlation spectra of unphosphorylated (black) and hyper-pSer5 (red) CTD2′. **(c)** Annotation of pSer5/pThr5 resonances in the downfield region of the 2D H-N correlation spectrum. **(d)** Proline region from the 2D C′-N correlation spectrum of unphosphorylated (black) and hyper-pSer5 (red) CTD2′ with some proline resonances annotated.

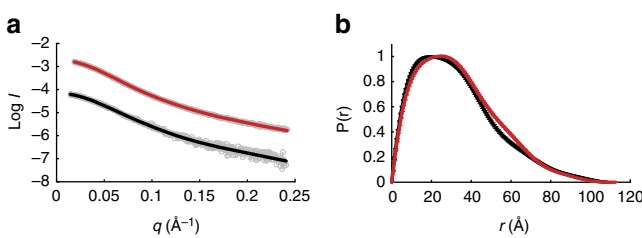

**Figure 4 | Small angle X-ray scattering reveals no significant change in pair-wise distances within CTD2′ upon extensive serine 5 phosphorylation. (a)** Raw scattering data for unphosphorylated CTD2′ (grey circles, bottom) and hyper-pSer5 CTD2′ (grey circles, top). Fits for unphosphorylated CTD2′ and hyper-pSer5 CTD2′ are shown superimposed on the raw data (solid black and red lines, respectively). **(b)** Representative pair-wise distance distributions for unphosphorylated CTD2′ (black) and hyper-pSer5 CTD2′ (red) calculated using the autoGNOM function in Primus qt, where the error bars represent the fit error.

19 proline residues (Fig. 5b, Supplementary Fig. 6c). Further, where isomerization occurred, peak splitting into two sets of NMR resonances was observed for Thr4, Ser7, Cys7 and Asn7 residues, accompanied by large chemical shift perturbations (blue bars, Fig. 2f). The presence of two sets of assignable resonances suggests a chemical exchange process on the millisecond timescale or slower. To confirm exchange between *cis*- and *trans*-proline isomers, we collected $^{15}$N ZZ-exchange NMR spectra, which permits the quantitative observation of conformational exchange on the ms–s timescale, in the presence of the *Drosophila* prolyl isomerase Dodo[29] (Supplementary Fig. 7). Exchange peaks were observed for all Pro6 residues adjacent to pSer5, but not for Pro3, consistent with Dodo specificity for the pSer5–Pro6 pair[30]. Interestingly, no exchange could be observed for the pThr5–Pro6 pair in the YTPVpTPS sequence context, suggesting that this heptad is essentially *trans*-locked on the 100 ms timescale, even in the presence of a prolyl isomerase. Thus, in the hyper-pSer5 state, CTD2′ experiences slow exchange between *trans*- and *cis*-proline isomers. In this exchange regime, peak intensities correspond to the populations of *cis*- and *trans*-proline species, which allowed

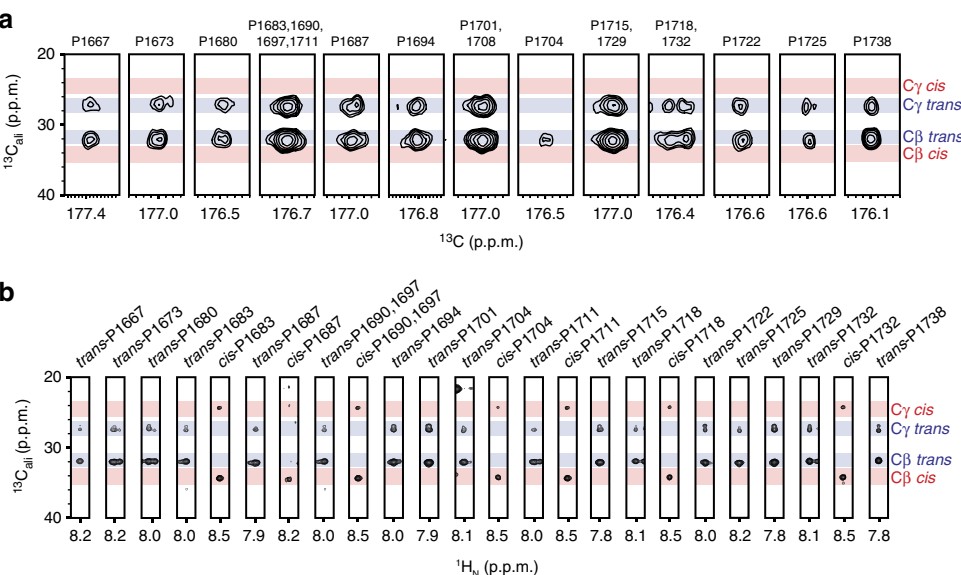

**Figure 5 | Structural characterization of the unphosphorylated and pSer5 CTD2′ by NMR spectroscopy.** (**a**) Cβ and Cγ chemical shifts from the 3D CCCON spectrum of unphosphorylated CTD2′ demonstrate that, when resolved, individual proline side chain resonances show a nearly all-*trans* state. Blue and red bars represent the range of chemical shifts (mean ± s.d.) for prolines in the *trans* and *cis* conformation, respectively. (**b**) Cβ and Cγ chemical shifts from the 3D CCCONH spectrum of hyper-pSer5 CTD2′ reveal dramatic *trans* to *cis* conformational switches in response to pSer5.

us to estimate the magnitude of *cis*-proline within each heptad (Fig. 6a). Within repeats of YSPTpSPS and the similar cysteine-containing repeat of YSPTpSPC, *cis*-Pro6 content was enriched threefold (to ~15%) by Ser5 phosphorylation. Further, Pro3 within all heptads containing pSer5 showed a modest enrichment of *cis*-proline content by ~5%, suggesting some non-local effects. Strikingly, Pro6 within heptads containing Asn7 (YSPTpSPN) showed a sixfold (~35%) enrichment in *cis*-proline. Thus the proline *trans*-to-*cis* switch is sequence context-dependent and modulated by both phosphorylation and deviations from the consensus heptad sequence.

**Heptad-specific proline switches modulate Ssu72 activity**. The observation that deviations from the consensus heptad sequence modulate the extent to which Pro6 *cis*–*trans* equilibria are affected by pSer5 suggests that, in response to uniform phosphorylation patterns, non-consensus heptads may impart an additional layer of specificity for CTD interacting factors. The CTD phosphatase Ssu72 has been shown to exhibit activity towards heptads containing the pSer5–*cis*Pro6 dipeptide pair[31]. Thus we hypothesized that non-consensus heptads within the hyper-pSer5 CTD modulate the apparent Ssu72 activity through pSer5 induced Pro6 isomerization. To test this possibility, we followed the dephosphorylation of hyper-pSer5 CTD2′ by *D. melanogaster* Ssu72-symplekin using RT-NMR spectroscopy (Fig. 7). Loss in NMR peak intensities relative to the zero time point were observed for all heptads containing pSer5–Pro6 dipeptide pairs (Fig. 7a, Supplementary Fig. 8). Interestingly, in each heptad sequence context, different apparent rates of dephosphorylation were observed (Fig. 7b, Supplementary Table 4). For pSer5 residues within the region flanked by the two consensus heptads, similar apparent rates were observed, suggesting that small deviations from the consensus motif (YSPTpSPC or YSPSpSPS) do not dramatically alter Ssu72 activity. However, minimal Ssu72 activity was observed for pS1682 (YSPNpSPS) and no dephosphorylation could be observed for pS1675 (YSPSpSSN), consistent with the requirements of Thr4 and Pro6 for Ssu72 activity[31,32].

Strikingly, Ssu72 exhibited nearly threefold apparent activity enhancement towards pSer5 residues within the Asn7 heptads, relative to the consensus motifs, strongly suggesting that the higher propensity for *cis*-Pro6 within these motifs increases the apparent Ssu72 activity.

To understand how residues flanking the pSer5–Pro6 pair affect pSer5 dephosphorylation by Ssu72, we analysed the conformation of phosphoryl CTD peptides upon Ssu72 binding. Several structures have been published for *Drosophila* or human Ssu72 bound to CTD peptides of different phosphorylation states, including *Drosophila* Ssu72, *Drosophila* Ssu72-Symplekin and human Ssu72-Symplekin, bound to pSer5 CTD (PDB codes: 3P9Y[16], 4IMJ[33] and 3O2Q[34], respectively), and *Drosophila* Ssu72-symplekin bound to a CTD peptide with Thr4/Ser5 doubly phosphorylated (PDB code: 4IMI[33]). The superimposition of these structures reveals that all known phosphoryl CTD peptides adopt a tight turn facilitated by *cis*-proline upon binding to Ssu72-symplekin (Fig. 8a). This tight turn is stabilized by three intra-molecular hydrogen bonds formed by the hydroxyl side chain of Thr4 with the main chain carboxylate of Ser7 (2.8 Å) and the amide group of Tyr1 in the following repeat (3.3 Å), as well as the main chain carboxylate of Thr4 and Pro6 (3.2 Å). Due to the high conservation of the tight turn configuration, the identity of the residues flanking Pro6 can be altered for effective Ssu72 recognition as long as the intra-molecular hydrogen bond network is maintained. For example, our NMR results demonstrate that Ssu72 dephosphorylates YSPSpSPS or YSPTpSPC with similar efficacy as it does consensus heptads; molecular modelling suggests that, in all three of these heptad motifs, the intra-molecular hydrogen bond network can be conserved even as the sequence varies (Fig. 8b). On the other hand, little to no Ssu72 activity was observed upon the replacement of Thr4 by Asn. We attribute this loss of activity to the need for Asn4 to adopt an alternative rotameric state to avoid steric clashes, which is incompatible with forming two of the hydrogen bonds that stabilize the Ssu72 recognition conformation (Fig. 8c). We have shown previously that the phosphorylation of Thr4, which also

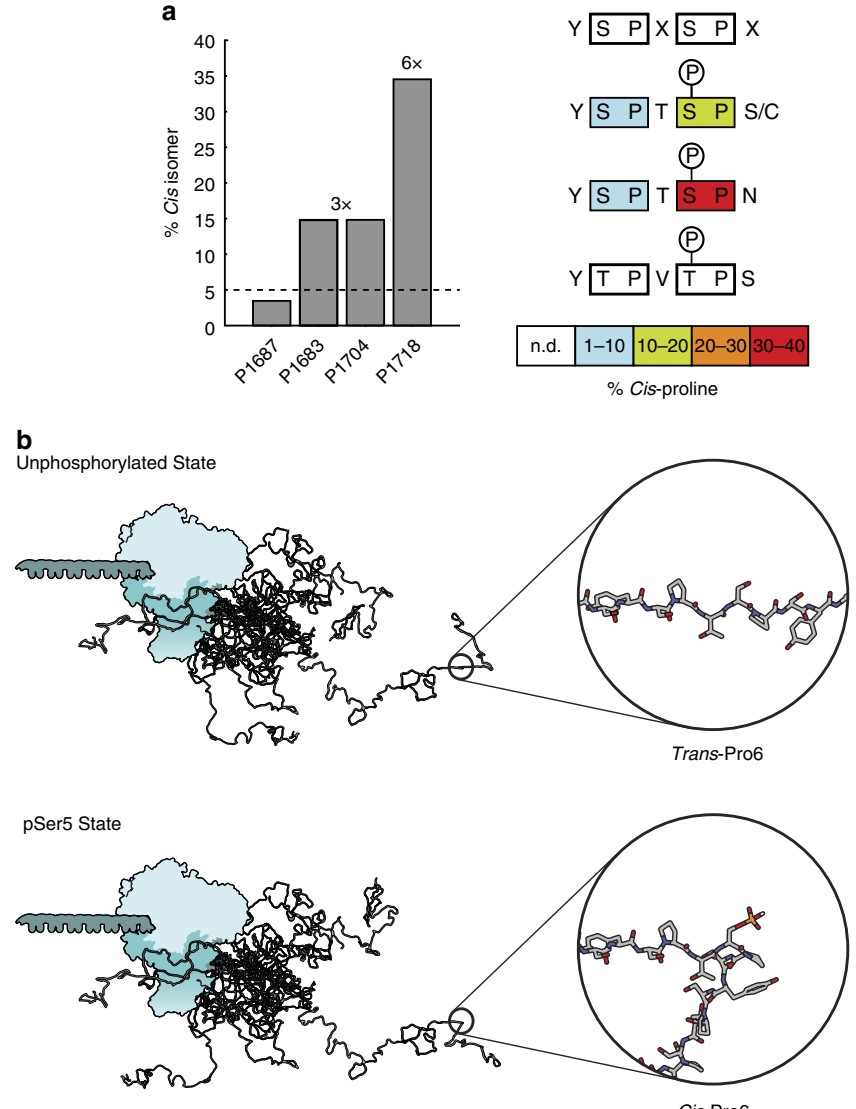

**Figure 6 | The impact of serine 5 phosphorylation on the structure of the Dm CTD.** (**a**) Percentage of *cis*-proline for several proline residues determined from peak intensities in 2D NMR correlation spectra of hyper-pSer5 CTD2′, where the dotted line denotes the average percentage of *cis*-proline in the unphosphorylated state (left). This is depicted schematically for various heptad sequences in CTD2′ (right). (**b**) Model for the effect of Ser5 phosphorylation on the structure of the CTD. In the unphosphorylated state, the CTD exists in an ensemble of conformational states that favour prolines in the *trans* conformation (top). Hyper-pSer5 incorporation causes the CTD heptad repeats bearing the sequence motifs highlighted in **a** to undergo dramatic structural rearrangement driven by pSer5-dependent proline isomerization.

disrupts these two hydrogen bonds, reduces Ssu72 activity fourfold[33].

In the context of the present study, the most significant observation of our NMR analysis of Ssu72-catalysed CTD2′ dephosphorylation is that Ssu72 shows its greatest activity toward pSer5-CTD heptads containing Asn7, which we have also shown are the heptads most highly enriched in *cis*-proline among those observed in this region of the CTD. For this heptad, molecular modelling suggests that the side chain of Asn7 could form an additional intra-molecular hydrogen bond with the carboxylate group of Thr4 (Fig. 8d). With all possible isomeric states of the Asn sidechain, the most favourable configuration is within 3.2 Å away from the backbone carboxylate of Thr4, which further strengthens the tight turn conformation needed for Ssu72 recognition and makes the *cis*-Pro6 more energetically favourable. Taken together, this data strongly suggest that, in the presence of uniform phosphorylation patterns, non-consensus CTD heptads

encode cryptic structural switches to fine tune the specificity of CTD interacting factors.

## Discussion

Based on these results, we propose a model in which hyper-pSer5 does little to alter the scaling properties of the CTD2′ region of *Drosophila* CTD; specifically the measured $R_g$ and $D_{max}$ are unaltered. Instead, dramatic structural rearrangements occur on the single heptad scale, driven by sequence context-dependent proline *trans*-to-*cis* isomerizations (Fig. 6b). A general conclusion from these observations is that these features allow the CTD to transduce homogenous PTMs into structurally and functionally diverse responses. This discovery predicts multiple potential mechanistic outcomes in the context of the CTD code.

The first general conclusion supported by our findings is that sequence context-dependent structural switches created through

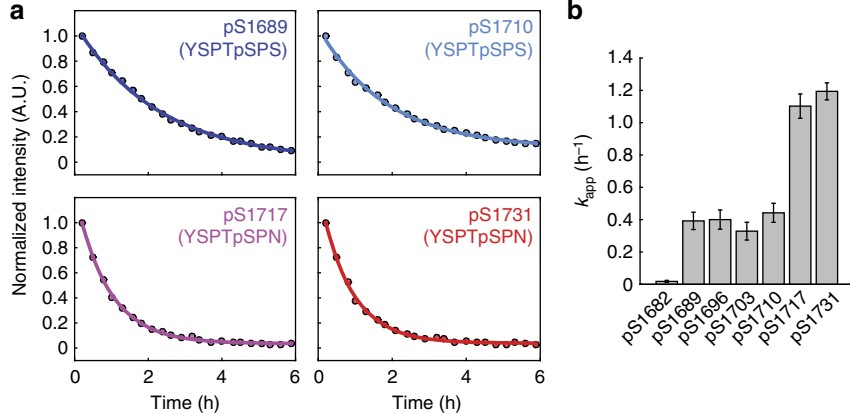

**Figure 7 | Structural switches in hyper-pSer5 CTD2′ modulate the apparent Ssu72 activity.** (**a**) Representative kinetic traces of Ssu72 dephosphorylation of pSer5 in CTD2′ monitored by RT-NMR. (**b**) Apparent rate constants for pSer5 dephosphorylation reveal heptad-specific Ssu72 activities. The highest apparent Ssu72 activities are observed for pSer5 residues within heptads containing Asn7. Error bars represent the errors from non-linear least squares fitting. All fitting procedures are described in detail in Methods section.

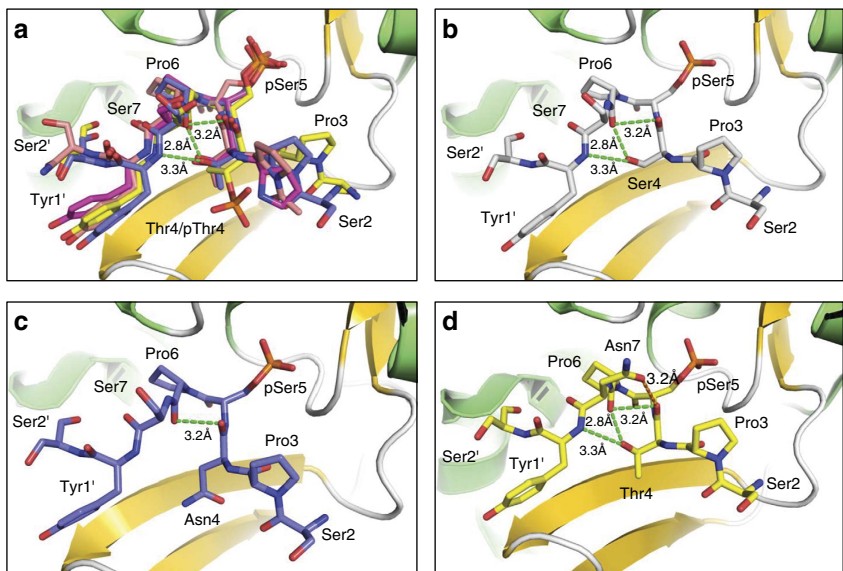

**Figure 8 | The conserved conformation of CTD peptides recognized by Ssu72.** (**a**) Ssu72 is shown as a ribbon diagram with α-helices in green and β-strands in gold. CTD peptides are shown as coloured sticks with carbon atoms shown in different colours: PDB code 4IMI (yellow), 4IMJ (blue), 3P9Y (salmon), and 3O2Q (magenta). The intra-molecular hydrogen bonds are shown in green dashed lines. The CTD residues are numbered based on consensus sequence and the following repeat residues are labelled with a prime. (**b**) The intra-molecular hydrogen bond network can be maintained even when Thr4 is replaced by Ser. (**c**) The replacement of Thr4 by Asn loses two intra-molecular hydrogen bonds. (**d**) An additional intra-molecular hydrogen bond can be formed (orange dashed line) when Ser7 is replaced by Asn.

enriched *cis*-proline isomerization have the ability to facilitate or impair the binding of isomer-sensitive CTD interacting factors at specific regions of the CTD. The diversity of CTD sequences across eukaryotes has been recently acknowledged[3], though the functional significance of many non-consensus heptads has not been widely investigated. Non-consensus CTD repeats may expand the repertoire of available PTMs, thus increasing the complexity of signalling through the CTD[4,35–38]. In our minimal system, we observed that non-consensus motifs that contained conservative deviations from the consensus responded similarly to Dm P-TEFb and Ssu72. In contrast, more cryptic variants such as YSPNSPS produced drastically altered outcomes, depending on the modifying enzyme present, and substantial deviation from the consensus heptad sequence rendered some repeats resistant to modification by both enzymes. Even in the context of the heptads

that conform the least well to the consensus, we emphasize that our assays did not yield substantial phosphorylation of Ser2 residues, suggesting a strong specificity of Dm P-TEFb for serine residues occupying the 5-position of the heptad *in vitro*.

Our findings suggest that the relationship between consensus conservation and functional specialization in the CTD may lie on a continuum. In this view, conservative sequence deviations may be tolerated by the majority of regulating enzymes, imparting only modest differences in modification kinetics and patterning. By contrast, more dramatic deviations from the consensus sequence may only support interaction with a subset of regulatory factors. While the unique functions the full set of conserved non-consensus motifs serve during transcription will need to be determined empirically, our first set of observations provide support for the emerging hypothesis that variation in CTD

sequence enables differential gene regulation in the context of normal development or at the level of individual genes[35,36]. In this context, our model leads to the prediction that conserved heptads maintain the ability to attract a wide range of factors involved in basic cellular processes, while non-consensus heptads enhance spatial control of interacting factor recruitment, thus creating more tightly regulated transcriptional programmes in higher eukaryotes.

Our observation that non-consensus heptads within the Dm CTD-containing Asn7 show a preponderance for *cis*-pro6 in response to pSer5 is striking. These heptads are conserved from yeast to human and tend to cluster in a region just beyond the consensus repeats[7]. For example, the human CTD has five heptads that contain Asn7, clustered between repeats 20 and 30. Our results suggest that, as in *Drosophila*, these regions should populate a high degree of *cis*-proline when pSer5 marks are prevalent, as in early phases of the transcription cycle. This could restrict the binding of many known pSer5-*trans*-Pro6 CTD interacting factors or, alternatively, favour the association of pSer5-*cis*-Pro6-specific factors. Ssu72 is a pSer5-*cis*-Pro6-specific phosphatase known to promote the transition from hyper-pSer5 to hyper-pSer2 during transcript elongation[31]. In line with this model, our results demonstrate that Asn7 heptads can increase the apparent Ssu72 activity targeted to a cluster of heptads in the CTD. Recent work has revealed that most CTD heptads are phosphorylated only once at any given time[9,10], suggesting ordered erasing of the pSer5 mark prior to writing of the pSer2 mark. Further, the average number of phosphates carried by yeast and human CTDs appears to be much less than the number of repeats, underpinning the transient nature of CTD modifications. Our results suggest that one role of Asn7 repeats may be to pre-prime this particular region of the CTD for pSer5 dephosphorylation by Ssu72, thus assisting in the dynamic spatiotemporal control of the pSer5-to-pSer2 transition.

Finally, the observations we report here predict a potential mechanistic route to the establishment of rare phospho-marks. For example, it has been shown that certain CTD interacting factors, such as the histone methyltransferase Set2 recognize doubly phosphorylated repeats[1]. Notably, we observe little to no Ssu72 activity on YSPNSPS and YSPSSSN heptads phosphorylated on Ser5. Thus these heptads remained poised for Ser2/Ser5 double phosphorylation, which may ultimately recruit specific regulartory enzymes to the CTD, such as Set2. Such a gradient in dephosphorylation following distributive phosphorylation could generate multiple epitopes for obligate co-transcriptional processes (such as capping enzyme binding pSer5 repeats) while ensuring a minimum number of epitopes are created for processes that may not be essential for each round of transcription (for example, Set2-mediated histone methylation in gene bodies). Thus the strategy employed here highlights the general feasibility of applying quantitative biophysical techniques to obtain mechanistic insights into the CTD code, revealing a multi-layered mechanism for the regulation of spatially and temporally ordered recruitment of factors to the CTD.

## Methods

**Fly procedures.** Sequences encoding RNAi-resistant Rpb1[WT] or Rpb1 derivatives with a double FLAG-tag at the C-terminus were subcloned into the pUASt-attB vector, followed by transformation into the attP site on chromosome 3 in the *PhiC31 attP 86Fb* fly line[39]. *UAS-Rpb1i* and *yw; Actin-GAL4/CyO* were obtained from the Bloomington Stock Center (lines 36,830 and 4,414, respectively). Rpb1i resistance of the ectopically expressed Rpb1 variants was achieved by changing the part of the coding sequence of Rpb1 that corresponds to the 21 nt RNAi recognition sequence (sense strand: AACGGTGAAACTGTCGAACAA) to AACCGTCAAGTTGAGCAACAA. The *UAS-Rpb1i, UAS-Rpb1* lines were generated by routine matings and meiotic recombination. The lethality test was done by mating virgin female *yw; Actin-GAL4/CyO* to male *yw; UAS-Rpb1i, UAS-Rpb1[WT/mut]*. Animals were raised at 21 °C. Rescue was confirmed by the emergence of straight-winged adults among the progeny (*Actin-GAL4/+; UAS-Rpb1i, UAS-Rpb1[WT/mut]/+*).

Western blot analysis for ectopic expression of Rpb1 was done by dissecting late pupae from the pupal case and then homogenizing and boiling the tissue in LDS sample buffer (Invitrogen). Equal numbers of male and female late pupae were selected and pupae of the genotype *yw; Actin-GAL4/+; UAS-Rpb1i, UAS-Rpb1[WT/mut]/+* were distinguished from the *yw; CyO/+; UAS-Rpb1i, UAS-Rpb1[WT/mut]/+* counterpart by the intensity of red pigment in the eye (the *UAS-Rpb1* and *Actin-GAL4* transgenes have white gene markers). For western blotting, tissues equivalent to 0.3 pupae were loaded into each lane on a 3–8% Tris-acetate SDS–polyacrylamide gel electrophoresis (PAGE) gel (Life Technologies). And a broad range (11–245 kDa) protein ladder (p7712 NEB) was run to provide molecular weight markers. Transgenic Rpb1 expression was detected with rabbit anti-Flag antibody (1:3,000; Genscript). Spt5 was detected with rabbit anti-Spt5 (1:3,000). The blot was subsequently probed with goat anti-rabbit IgG (1:3,000; Alexa Fluor 488) and visualized with a Typhoon (GE Healthcare).

**Protein expression and purification.** *Carboxy-terminal domain 2'.* A synthetic gene for the *Drosophila melanogaster* RPB1 Carboxy-terminal domain was purchased from GeneArt (Thermo Fisher Scientific). A region corresponding to residues (1,657–1,739) was amplified by PCR, cut with XhoI (NEB) and XmaI (NEB) and ligated into the pET49b+ expression vector (Novagen) using T4 DNA ligase (NEB) to produce a construct containing glutathione *S*-transferase and His tags. Protein expression was performed in *E. coli* BL21 DE3 cells. Batch cultures (500 ml) were grown at 37 °C in Luria–Bertani (LB) medium supplemented with 30 μg ml$^{-1}$ kanamycin. At an optical density at 600 nm (OD600) of 0.8, cells were induced using 0.5 mM isopropyl-β-D-thiogalactopyranoside (IPTG) and allowed to incubate at 37 °C for 3 h. Following lysis by sonication on ice in lysis buffer (50 mM Tris/HCl pH 7.5, 500 mM NaCl, 20 mM Imidaozole, 2.5 mM β-mercaptoethanol, 10× EDTA-free protease inhibitor cocktail (Calbiochem) and 10 units of RNAse free DNAse (NEB)), samples were centrifuged at 4 °C for 40 min at 11,500 *g*. Cleared supernatant was passed over HisPur Ni$^{2+}$-NTA resin (Thermo Fisher Scientific) and bound protein was washed of contaminants using 5 column volumes of wash buffer (50 mM Tris/HCl pH 7.5, 500 mM NaCl, 20 mM imidaozole, 0.1% Triton-1000, 2.5 mM β-mercaptoethanol). Protein was eluted using elution buffer (50 mM Tris/HCl pH 7.5, 500 mM NaCl, 200 mM imidaozole, 2.5 mM β-mercaptoethanol). The glutathione *S*-transferase and 6× His tags were removed by adding recombinant His-tagged HRV 3C protease to the protein (resulting in an N-terminal non-native GPG) and dialysing the mixture overnight against 50 mM Tris/HCl pH 7.5, 300 mM NaCl, 2.5 mM β-mercaptoethanol at 4 °C. The protein was then passed over the Ni$^{2+}$-NTA column to remove the protease and non-specifically bound contaminants. A final purification was then performed by size exclusion chromatography in 80 mM Imidazole pH 6.5, 50 mM KCl and 2.5 mM β-mercaptoethanol using P-10 resin (BioRad).

*Dm P-TEFb.* Sf9 cells were grown in suspension at 27 °C to 1.5 million cells ml$^{-1}$ and infected with 1/10 culture volume *D. mel* P-TEFb virus (generous gift from J.T. Lis). Infection was carried out at 27 °C at a shaker speed of 75 r.p.m. for 72 h. Following lysis in 50 mM HEPES pH 7.5, 500 mM NaCl, 10% glycerol, 1% Nonidet P-40, 2.5 mM imidizole and 2.5 mM β-mercaptoethanol and protease inhibitors, by dounce homogenization, lysates were centrifuged at 4 °C for 30 min at 100,000 *g*. Cleared supernatant was passed over TALON resin (Clontech) and bound protein was washed using 5 column volumes of 50 mM HEPES pH 7.5, 500 mM NaCl, 10% glycerol, 1% Nonidet P-40, 10 mM imidazole and 2.5 mM β-mercaptoethanol. Dm P-TEFb was eluted with 50 mM HEPES pH 7.5, 500 mM NaCl, 10% glycerol, 1% Nonidet P-40, 200 mM imidazole and 2.5 mM β-mercaptoethanol and flash frozen. Kinase activity towards CTD2' was confirmed by auto-radiography.

*Dm Dodo.* A synthetic gene corresponding to residues (1–166) of *D. mel* Dodo was purchased from GeneArt (Thermo Fisher Scientific). The gene was cut with XhoI (NEB) and XmaI (NEB) and ligated into the pET47b+ expression vector (Novagen) using T4 DNA ligase (NEB) to produce a His-tagged construct. Protein expression was performed in *E. coli* BL21 DE3 cells. Batch cultures (500 ml) were grown at 37 °C in LB medium supplemented with 30 μg ml$^{-1}$ kanamycin. At an OD600 of 0.8, cells were induced using 0.5 mM IPTG and allowed to incubate at 37 °C for 3 h. Following lysis by sonication, samples were centrifuged at 4 °C for 40 min at 11,500 *g* and purified by affinity chromatography using HisPur Ni$^{2+}$-NTA resin (Thermo Fisher Scientific) and eluted using elution buffer. Following removal of the His-tag using His-tagged HRV 3C protease, a final purification was performed by gel filtration in 50 mM Tris/HCl pH 7.5, 150 mM NaCl, 1 mM EDTA and 2.5 mM β-mercaptoethanol using a sephacryl S-100 Hi-prep 16/60 size exclusion column on an Äkta FPLC (GE).

*Dm Ssu72 phosphatase and symplekin.* Full-length *D. mel* Ssu72 phosphatase (residues 1–195) was cloned into the pET28b derivative vector. The final Ssu72 construct contained an N-terminal 8× His-tag and SUMO tag followed by a PreScission protease site. *D. mel* symplekin (residues 19 − 351) was also cloned into a pET28b derivative vector. Both proteins, Ssu72 and Symplekin, were purified individually as follows. The plasmids were transformed into *E. coli* BL21 (DE3) cells, which were grown in 1 litre of LB medium supplemented with 50 μg ml$^{-1}$ kanamycin. Upon culture growth to an OD600 of 0.4–0.6, expression was induced by adding IPTG to a final concentration of 0.5 mM and the cultures were grown at

16 °C for an additional 16 h. The cultures were then pelleted, lysed by sonication (lysis buffer containing 50 mM Tris pH 8.0, 500 mM NaCl, 10 mM Imidazole, 0.1% Triton X-100 and 10% glycerol) and centrifuged to separate cell debris. Each protein was purified individually by running the aqueous fraction through a Ni-NTA column (Qiagen) and eluting with imidazole. The tags were removed by thrombin cleavage during dialysis at 4 °C overnight. Each protein was further purified by gel filtration (Superdex-75 GE Healthcare) and concentrated. Purity was verified by SDS–PAGE through each step.

To form the Ssu72-symplekin complex, the proteins were incubated at a molar ratio of 1:1.5 (Ssu72:symplekin) overnight at 4 °C in a dialysis bag. The complex was separated from unbound protein by running through gel filtration (Superdex-200 GE Healthcare). And complex formation was verified via SDS–PAGE. Complex fractions were pooled and concentrated to ~6.8 mg ml$^{-1}$. The complex was stored in buffer of 25 mM Tris-HCl at pH 8.0, 200 mM NaCl and 1 mM DTT and flash frozen to $-80$ °C.

**Kinase reactions.** Kinase reactions were generally carried out in 50 mM Tris/HCl pH 7.5, 50 mM NaCl, 10 mM MgCl$_2$, 2 mM DTT, 12 mM ATP with 80 μg ml$^{-1}$ Dm P-TEFb and 100 μM CTD2′. Reactions were carried out at 24 °C and allowed to proceed to completion (~16 h), after which 20 mM EDTA was added to ensure reaction termination.

**Mass spectrometry.** *Matrix-assisted laser desorption/ionization–time of flight (MALDI-TOF) MS.* MALDI TOF mass spectra of intact unphosphorylated CTD2′ and phospho CTD2′ were acquired on an Ultraflextreme instrument (Bruker, Billerica, MA) in linear positive detection mode using factory-configured instrument parameters for 5,000 − 20,000 $m/z$ range. The instrument was calibrated using a protein mixture containing bovine insulin, MW 5733.5; bovine ubiquitin, MW 8564.8; bovine RNAse A, MW 13682.2; equine heart cytochrome C, MW 12359.9 and equine heart myoglobin, MW 16951.3 (all from Sigma); a 20 mg ml$^{-1}$ solution of super-DHB (2,5-dihydroxybenzoic acid and 2-hydroxy-5-methoxybenzoic acid, Sigma) in 50% aqueous acetonitrile (ACN) containing 0.1% o-Phosphoric acid (EMD Millipore) and 0.1% trifluoroacetic acid (Thermofisher) was used as the matrix for both the calibrants and the CTD samples. The CTD samples for the MALDI TOF MS were prepared by mixing 1 μl of 100 μM protein solution in water and 1 μl of the matrix solution and applying 1 μl of this mixture to a polished stainless steel target. Mass spectra were acquired by summing 2,000–3,000 shots at a 1,000 Hz laser repetition rate, average calibration error was 521 p.p.m.

*Chymotrypsin digest.* A 1 μg μl$^{-1}$ stock solution of sequencing-grade chymotrypsin (Thermofisher) prepared in 1 mM hydrochloric acid was diluted 50-fold with chilled 50 mM aqueous solution of triethylammonium bicarbonate and added to the CTD solution in a 1:1 vol:vol ratio. The proteolysis was allowed to proceed overnight at 37 °C. Samples were acidified with a 1% aqueous solution of formic acid (FA), dried down and re-dissolved in 15 μl of 4% ACN containing 0.1% FA for the nano-LC MS$^2$ analysis.

*Nano-LC MS$^2$.* Digested peptide solution (3 μl) was loaded onto an Acclaim PepMap100 trapping column (100 μm × 2 cm, C18, 5 μm, 100 Å, Thermo) at a flow rate of 20 μl min$^{-1}$ using 4% aqueous ACN, 0.1% FA as a mobile phase. The peptides were separated on an Acclaim PepMap RSLC column (75 μm × 15 cm, C18, 2 μm, 100 Å, Thermo) with a 90-min 4–60% linear gradient of aqueous acetonitrile containing 0.1% FA. The gradient was delivered by a Dionex Ultimate 3,000 nano-LC system (Thermo) at 300 nl min$^{-1}$.

An LTQ Orbitrap Velos electron transfer dissociation (ETD) mass spectrometer (Thermo) was set to acquire data using the following data-dependent parameters. A full FT MS scan at $R$ 60,000 over 350–2,000 $m/z$ range was followed by 5 FT MS$^2$ scans with CID activation and 5 FT MS$^2$ scans with ETD activation on most intense precursors at $R$ 7,500. Only the precursors with charge states $+2$ and higher were selected for MS$^2$ based on the FT master scan preview; the charge-state-dependent ETD time and monoisotopic precursor selection were enabled; the isolation window was 5 $m/z$, and the minimum precursor signal was set at 10,000 counts for both CID and ETD. The ETD activation time was 100 ms. Polysiloxane ion, $m/z$ 445.12003 was used as a lock mass.

*Data analysis.* The mass spectra were processed using Proteome Discoverer 1.3 (P.D. 1.3, Thermo). The expressed CTD sequence was appended to a database containing 45,443 sequences (*D. Melanogaster*, *E. Coli*, common contaminants). The workflow was split into pipelines for CID and for ETD; and the following search parameters were used for both pipelines: enzyme chymotrypsin with 3 missed cleavages, precursor mass tolerance 30 p.p.m., fragment mass tolerance 0.8 Da, Met oxidation and Ser, Thr and Tyr phosphorylation as dynamic modifications, and Cys carbamidomethylation as static modification. The CID ion series weights for $b$ and $y$ were set to 1; and the ETD ion series weights were 1 for $c$ and $z$ ions and 0.25 for $b$ and $y$ ions. The ETD pipeline included a non-fragment filter node, which was set to remove precursor peak, charge-reduced precursor peaks and peaks due to the neutral loss from charge-reduced precursors prior to the SEQUEST search. Resulting search files were exported into Scaffold PTM (Proteome Software) where phospho-site probabilities were determined using the Ascore algorithm[40].

**NMR spectroscopy.** For NMR experiments, expression of CTD2′ was performed in M9 minimal media enriched with $^{15}$N-NH$_4$Cl and/or $^{13}$C-D-Glucose (Cambridge Isotope Laboratories). Following purification, samples were buffer exchanged in Amicon Ultra-15 3,000 NMWL centrifugal filters (Merck Millipore Ltd.). Typically, 80 mM imidazole pH 6.5, 50 mM KCl, 10% glycerol, 2 mM DTT and 10% D$_2$O was used for NMR experiments. For $^{15}$N ZZ-exchange experiments, 20 mM MES pH 6.5, 50 mM KCl, 10% glycerol, 2 mM DTT and 10% D$_2$O was used. To obtain the desired pH range during $^{31}$P experiments, 80 mM Citrate pH 4.0/5.0/5.5, 80 mM imidazole pH 6.2/6.5 and 80 mM Tris/HCl pH 7.2/8.3 containing 50 mM KCl, 10% glycerol, 2 mM DTT and 10% D$_2$O were used.

NMR Spectra were collected at the Lloyd Jackman NMR facility at the Pennsylvania State University on Bruker Avance-III spectrometers operating at proton frequencies of 500, 600 or 850 MHz equipped with TCI single-axis gradient cryoprobes ($^1$H/$^{13}$C/$^{15}$N/$^2$H) with enhanced sensitivity for $^1$H and $^{13}$C. Phosphorous-detect experiments were performed on a 500 MHz Bruker Avance-III-HD spectrometer equipped with a broadband (BBO) Prodigy CryoProbe. Chemical shift assignments were made using $^{13}$C-Direct Detect methods developed in-house[17,41,42], as well as standard $^1$H-Detect triple resonance experiments. $^{13}$C and $^{31}$P chemical shifts were referenced to 4,4-dimethyl-4-silapentane-1-sulfonic acid (DSS) and phosphoric acid standards, respectively. 2D Spectra were processed in Topspin 3.2 (Bruker) and NMRPipe and analysed in Sparky. 1D spectra were processed in Topspin 3.2 (Bruker) and analysed in Mnova (Mestrelab Research). Average chemical shift perturbations for CTD2′ upon phosphorylation were calculated by:

$$\Delta\delta_{AV} = \left[ \frac{1}{3} \left\{ (\Delta\delta_{HN})^2 + (0.102\,\Delta\delta_N)^2 + (0.251\,\Delta\delta_{C'})^2 \right\} \right]^{1/2}$$

where $\Delta\delta_{HN}$, $\Delta\delta_N$ and $\Delta\delta_{C'}$ are the differences in $^1$H, $^{15}$N and $^{13}$C′ chemical shift between the unphosphorylated and phosphorylated species, respectively.

**RT-NMR kinetics and data processing.** Dm P-TEFb kinase reactions were performed in 50 mM HEPES pH 6.8, 50 mM KCl, 20 mM MgCl$_2$, 12 mM ATP, 2 mM DTT and 10% D$_2$O with 250 μM CTD2′ and ~100 μg ml$^{-1}$ Dm P-TEFb. Standard $^1$H, $^{15}$N-HSQC spectra were collected at 850 MHz with 1,024(H) × 256(N) points, 4 scans and a recycle delay of 0.8 s for total acquisition times of ~16 min for the first 8 h, after which 16 scans were collected. To measure slow sites, 16 scans were acquired for each experiment over the entire time course. As recalibration of the instrument was required following enzyme addition, the first data point was acquired in an effective dead time of ~20 min. Phosphatase reactions were performed in 80 mM imidazole pH 6.5, 50 mM KCl, 2 mM DTT and 10% D$_2$O with 1.0 mM hyper-pSer5 CTD2′ and ~10 μg of *D. mel* Ssu72-symplekin complex. For the phosphatase reactions, spectra were collected as described using four scans throughout the time course. Spectra were processed in Mnova (Mestrelab Research). Extracted peak intensities for pSer/pThr resonances and effected resonances of the neighbouring residues were plotted as a function of time and fit in MATLAB. Single exponential decays and build-up curves were fit as irreversible first-order reactions and intermediate species were fit as two consecutive irreversible first-order reactions by the method of non-linear least squares using:

$$y = y_0 + S_0 e^{-k_{app} t}$$
$$y = y_0 + S_0 \left( 1 - e^{-k_{app} t} \right)$$
$$y = \frac{k_1 S_0}{k_2 - k_1} \left( e^{-k_1 t} - e^{-k_2 t} \right)$$

Where possible, $k_{app}$ was calculated as the average between a given pSer resonance and the resonances of neighbouring residues. Reported errors represent the 95% confidence intervals, or the propagation of error where $k_{app}$ represents an average.

**Small-angle X-ray scattering.** SAXS experiments for CTD2′ were performed in 80 mM Tris/HCl pH 7.5, 50 mM KCl, 10% glycerol and 5 mM DTT. SAXS data was collected at the Cornell High Energy Synchrotron Source (CHESS) on the G1 beamline. Incident radiation was produced at 9.963 keV with a flux of $8 \times 10^{11}$ photons s$^{-1}$ at 51 mA, providing a $q$-space range of 0.007–0.7 Å$^{-1}$. Scattering from a silver behenate standard was used for $q$-axis mapping. Data collection was performed using dual Pilatus 100K-S detectors. Reduction of the 2D images to 1D scattering profiles was performed using BioXtas Raw. Scattering profiles and uncertainties were computed as the average and s.d. of three exposures, with each exposure comprising 20 1-s frames. Solvent blanks were collected immediately before and after each protein sample exposure by measuring the scattering from the spin column flow through from each sample, and solvent subtraction was performed using equivalent numbers of frames. Data were collected at protein concentrations from 4–11 mg ml$^{-1}$ for both unphosphoryalted and pSer5 CTD2′. No signs of aggregation, inter-particle effects or radiation damage were observed. Average radius of gyration ($R_g$) values were determined for each sample using the Guinier approximation with $qR_g \leq 0.8$, as suggested for disordered systems, such as CTD2′ (ref. 43). Guinier fitting and pair-wise distance distribution calculations were performed using the method of non-linear least squares in MATLAB and the auto-GNOM function in Primus qt, respectively.

**Data availability.** The data that support the findings of this study are available from the corresponding author upon request.

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

## Acknowledgements

This work was funded by a US National Science Foundation grant to S.A.S. (MCB-1515974), NIH grant GM047477 to D.S.G., NIH grant GM104896 and Welch Foundation grant F-1778 to Y.J.Z., NIH training grant T32-AI074551 to B.P. and an MPHD scholarship from the Alfred P. Sloan Foundation to E.B.G. We thank the TRiP at Harvard Medical School (NIH GM084947) for providing the transgenic RNAi fly stock. The Cornell High Energy Synchrotron Source (CHESS) is supported by the National Science Foundation under awards DMR-1332208 and DMR-0936384; the Macromolecular Diffraction at CHESS (MacCHESS) facility is supported by the National Institutes of Health under award GM-103485.

## Author contributions

E.B.G. and T.N.L. designed experiments, performed experiments, analysed data and wrote the manuscript. F.L. designed experiments, performed experiments and analysed data. B.P., M.J.F. and B.P.M. designed experiments and performed experiments. Y.J.Z. analysed data and wrote the manuscript. D.S.G. and S.A.S. designed experiments, analysed data and wrote the manuscript.

## Additional information

**Competing interests:** The authors declare no competing financial interests.

