## [Peer Review File · Nature Communications]

Reviewers' Comments:

Reviewer #1 (Remarks to the Author):

The manuscript by Gibbs et al. describes the structural effects on the RNA polymerase II C-terminal domain induced by its phosphorylation. The authors use a broad spectrum of biophysical techniques to study the conformational changes in a CTD of 12 hepta repeats from *Drosophila melanogaster*. This amino acid sequence has a unique signature due to some deviations from the typical hepta repeat consensus YSPTSPS, which allows the full and specific NMR resonance peak assignment. Using NMR spectroscopy and SAXS the authors analyze the conformational changes that are induced upon the full (this means saturated) phosphorylation of this CTD stretch by the Dm kinase P-TEFb.

I am very enthusiastic about this study as I consider NMR spectroscopy the most unbiased method to analyze cis-trans conformational changes in prolines. Likewise, analysis of the chemical shift changes induced by Ser phosphorylation is the best means to identify the modified residues as also incomplete or partial modifications can be seen. Such study has been long awaited by the field and any analytical approach of CTD phosphorylation, e.g. by mass spectrometry, complements our view on this modification, which is still mostly derived from Western blot analysis. The study appears sound and the manuscript is very well written, only the discussion is short and offers not many ideas and implications to the general reader.

Comments and criticism:

It would be very much appreciated if the authors could add a western blot analysis of pSer2, pSer5 and pSer7 CTD phosphorylation to the study. Having such a profound analytical study that clearly shows Ser5 phosphorylation by Dm P-TEFb it would be a perfect link to the classical WB analysis, if the authors now also include how this phosphorylation pattern is regarded by the antibodies. This is meant as a validation of the antibodies and not of the analytical technique like NMR, but could add to the discussion about the promiscuity of the Abs.

Would the chemical shift changes upon serine phosphorylation, shown e.g. in Fig. 3c, allow for a kinetic analysis of the modification by integration of the peak intensity? In other words, could you identify if any residue is faster phosphorylated than the other one or are all residues phosphorylated in equal terms to saturation. Such insights could add to the discussion about a distributive or processive phosphorylation mechanism by the kinase and potentially even to the directionality of the phosphorylation reaction (if there is any). Such data would be similar to those shown in Fig. 7 for the phosphatase Ssu72 activity.

Fig. 2: I would appreciate if the authors use the same bar diagram display introduced in Fig. 1B to indicate which sequence of the Dm CTD is analyzed. This could be, e.g., above the detailed amino acids sequence in 2A.

Likewise: In the Supplemental Fig. 1 the authors show an alignment of five different CTDs. Besides indicating the highly conserved region with a red box, it would also be nice if they could indicate which stretch of the Dm CTD was used in this study, using the same annotation as in Fig. 1B. Also, please use the entire space up to the right for the sequence display (I was searching for the FAGSG... start of the CTD you analyzed).

Discussion: It is striking that the heptads in Dm CTD containing an Asn7 are most highly enriched in cis-proline conformations and thus a target for Ssu72. Human CTD has five repeats that contain N7, four of them are clustered between repeats 20 and 30. One could discuss this observation. Likewise, although several groups now reported that Cdk9 predominantly phosphorylate Ser5 of the CTD (also in yeast!), it might be worth mentioning that even under saturated phosphorylation levels (Fig. 2B) no Ser2 phosphorylation occurs. This also adds to the specificity of the kinase that does not phosphorylate any given Ser-Pro motif, meaning Ser2 and Ser5.

Fig. 2: I don't really understand the bar diagram in panel F. Gray bars indicate the pSer5 or pThr5 position. But why is the blue line that indicates the cis-proline enriched state then always two bars to the right from the gray bar? I expected the blue cis-Pro6 bar next to it.

Fig. 8D: In this model of an YSPTSPN repeat shouldn't the Ndelta2 nitrogen be the donor for the hydrogen bond with the backbone CO of Thr5 instead of the Odelta1?

Minor comments:

Lane 91: for clarity you could say: by human P-TEFb

Lane 129: The authors could mention here the median length of the pSer5-pSer5 distance, which seems to be about 18 Ang.

Lane 156: Please explain the ¹⁵N ZZ-exchange to the general reader.

Reviewer #2 (Remarks to the Author):

In this paper, the authors identify an eight heptad repeat-containing region of the RNA polymerase II rpb1 C-terminal domain that is essential for development in *Drosophila*. The

region is highly conserved among higher eukaryotes, suggesting that it is essential for development in other eukaryotes as well. This region is readily uniformly phosphorylated at each of the serine 5 positions of the heptad repeat by the enzyme P-TEFb, a result confirmed by mass spec and NMR. SAXS data indicate that the radius of gyration of this region is fairly large for the length of the region and phosphorylation does not significantly change the radius of gyration, despite the addition of greater than 20 negative charges. Hyperphosphorylation does increase the percentage of prolines that are in the cis conformation (by >3 fold for prolines adjacent to S5). Kinetic dephosphorylation experiments with the phosphatase Ssu72 indicate differential recognition of the phosphorylated heptad repeats by this phosphatase. The authors suggest that the heptad repeats form sequence dependent structural features that can be recognized by CTD interacting factors, like Ssu72. The authors further suggest that PTMs like phosphorylation are not read through their overall impact on the CTD structure, but on a heptad-by-heptad basis as part of a CTD-code.

1. The authors apply NMR and SAXS approaches and some limited sophisticated analyses, however they do not attempt to obtain structural models from their data. Instead, the authors write: “we analyzed the conformation of phosphoryl CTD peptides upon Ssu72 binding.” They use this Ssu-72 bound state to provide a concrete structural model rather than their own experimental data on the free non-phospho and phospho states. They also do not state how this structural analysis was performed; it should be indicated in the text. Importantly, the context of these structures should be provided (not just as PDB codes in the figure legend) and the citation should be indicated in the text. It is not clear how stable the heptad structural element is, though the narrow NH dispersion suggest that they are not stable. This will have a big impact on the model. Clearly, cis-trans proline isomerization occurs relatively slowly, but is still on the order of ms. Are the structural elements populated to a degree that hydrogen bonds or NOEs can be observed by NMR? Could this be performed using heptad repeat peptides? If the authors think these structural elements are not stable, this should be indicated in the text, since the reader will be heavily influenced by the structures shown in Fig. 8 in the Ssu72-bound state.
2. What are the implications of the change in the equilibrium populations of the cis-trans proline isomerization states? Are there different binding domains of various downstream partners for these two states? Could there also be a change in exchange rate, with the non-phosphorylated having some cis but in fast exchange?
3. While the phosphorylation does not have a significant impact on the radius of gyration, other properties of the CTD may be influenced. For example, there are several studies indicate that the CTD is involved in formation of hydrogels or liquid droplets. The authors seem to discount this hypothesis, but fail to address it experimentally. Within the context of the full CTD, does elimination of specific phosphorylation sites by mutation have the same effect as elimination of the 1659-1737 region. Does this match the expectations of the model?

4. The authors didn't push the phosphorylation reaction to obtain a more homogeneous/uniform sample; they obtain a mix, with a maximum of 10 sites. Newer published phosphorylation approaches such as those using ATP regeneration systems or dialysis methods could potentially yield almost uniform 12-site phosphorylation.
5. The finding that Ssu72 dephosphorylates the cis-stabilized sites is not novel since previous structural studies have already identified that aspect of the phosphatase.
6. The paper lacks excitement, not highlighting how the work is important to a deeper understanding of the function of the protein or the biological roles it plays. The discussion is only a single paragraph long, with little room to make connections between the findings in the paper and important biology.
7. A minor comment is that "apo" generally applies to a free state lacking a binding partner (protein, nucleic acid, small molecule ligand). As there are no binding partners studies, the authors' use of this term for the non-phosphorylated state is confusing.

Reviewer #3 (Remarks to the Author):

The manuscript by Gibbs et al addresses the difficult question of the structure of the RNA polymerase II CTD. The authors take a clever approach in taking advantage of the natural sequence variation present in the *Drosophila* CTD. They also used state of the art M/S and NMR spectroscopy. Their data reveal that the CTD (at least the fragment they analyzed) is intrinsically in an elongated form and that phosphorylation does not dramatically affects this overall structure. They also nicely show that serine phosphorylation affects the cis-trans state of the CTD prolines and that this is affected by the surrounding sequence. Using the phosphatase Ssu72 as an example, they showed how phosphorylation and proline isomerization can affect binding to the CTD and, more importantly, how sequence variation can impact on binding and activity of the phosphatase. This manuscript represents one of the very few successes in gaining some understanding on the structure of the CTD and on the impact of phosphorylation. It represents a small step forward in terms of new knowledge but, through its technical innovation and approach, may open the way to much more profound discoveries.

Reviewer 1

We thank Reviewer 1 for the helpful questions and comments that were provided to us and are glad that our manuscript was received well overall. Our responses to the reviewer's individual comments are provided below.

Comment 1: *It would be very much appreciated if the authors could add a western blot analysis of pSer2, pSer5 and pSer7 CTD phosphorylation to the study. Having such a profound analytical study that clearly shows Ser5 phosphorylation by Dm P-TEFb it would be a perfect link to the classical WB analysis, if the authors now also include how this phosphorylation pattern is regarded by the antibodies. This is meant as a validation of the antibodies and not of the analytical technique like NMR, but could add to the discussion about the promiscuity of the Abs.*

Response 1: We have attempted multiple western blots of our samples, using 8WG16 (intended epitope is unphosphorylated CTD), 4H8 and H14 (intended epitope is Ser5P), and H5 (intended epitope is Ser2P). All four antibodies were presented with both unphosphorylated and Dm P-TEFb treated Dm CTD. Unexpectedly, 8WG16 appears to have stained phospho-CTD more strongly than it stained unphosphorylated CTD. 4H8 and H14 only stained phospho-CTD and in both cases very poorly. This suggests that, being severely depleted in consensus heptads, Dm CTD is a poor target for these anti-Ser5P antibodies. No staining of either unphosphorylated or phosphorylated Dm CTD by H5 was observed. This is consistent with our MS and NMR observations, although the anti-Ser5P results suggest we cannot rule out the possibility that the Dm CTD sequence fails to display a recognizable epitope for this antibody. Given the paucity of conserved heptads in the Dm CTD, and the fact that the antibodies were raised against conserved heptads, we do not feel our western blot results allow for any definitive statements about the available antibodies to be made.

Comment 2: *Would the chemical shift changes upon serine phosphorylation, shown e.g. in Fig. 3c, allow for a kinetic analysis of the modification by integration of the peak intensity? In other words, could you identify if any residue is faster phosphorylated than the other one or are all residues phosphorylated in equal terms to saturation. Such insights could add to the discussion about a distributive or processive phosphorylation mechanism by the kinase and potentially even to the directionality of the phosphorylation reaction (if there is any). Such data would be similar to those shown in Fig. 7 for the phosphatase Ssu72 activity.*

Response 2: We thank the reviewer for providing us with the final motivation to do an experiment that we have had on our "to do" list ever since we performed the real-time NMR (RT-NMR) analysis of dephosphorylation. The reviewer is correct; after a methodical search, it was possible to find conditions under which we were able to monitor Dm CTD phosphorylation by Dm P-TEFb using RT-NMR. As the results were exciting, we are very glad to have done so. In brief summary of our new findings, phosphorylation appears to occur through a distributive mechanism, with rates of phosphate addition falling into three classes. All internal heptads containing a Ser5-Pro6 pair were well-phosphorylated and displayed essentially identical apparent rates. Representative traces are displayed in Figure 2E for consensus and Asn7-containing heptads (red traces). All internal heptads that lack a Ser5-Pro6 pair, and yet were phosphorylated, display lower apparent rate constants and fail to reach full saturation. A representative trace is displayed in Figure 2E in dark blue. Finally, the heptads at the artificial termini of our biophysical construct are very poor substrates, as represented by the light blue trace in Figure 2E. Figure S2 provides additional reporting on this experiment. The text has been modified as follows:

Real-time NMR (RT-NMR) permitted the kinetic measurement of CTD Δ 2' phosphorylation, revealing that for the seven internal heptads containing Ser5-Pro6 pairs, phosphorylation proceeded at similar apparent rates and reached comparable levels upon saturation (90%) (Fig. 2E, S2, Table S1) , with incomplete phosphorylation of the C-terminal repeat, and two internal repeats lacking a Ser5-Pro6 pair (Ser5 in YSPSSSN and Thr5 in YTPVTPS). The observed rates are consistent with a distributive mechanism similar to that observed for the human P-TEFb¹². Thus, the *in vitro* phosphorylation reactions produce a nearly complete hyper-pSer5 state, in agreement with our MS data (Fig. 2A-C,F).

Significantly, the results of the RT-NMR phosphorylation study provide a strong rationale for the distribution of phosphorylation states observed in the MALDI-MS experiment (Figure 2B), as will be discussed below in response to Reviewer #2, Comment 8.

Comment 3: *Fig. 2: I would appreciate if the authors use the same bar diagram display introduced in Fig. 1B to indicate which sequence of the Dm CTD is analyzed. This could be, e.g., above the detailed amino acids sequence in 2A.*

Response 3: The suggested modification has been made.

Comment 4: *Likewise: In the Supplemental Fig. 1 the authors show an alignment of five different CTDs. Besides indicating the highly conserved region with a red box, it would also be nice if they could indicate which stretch of the Dm CTD was used in this study, using the same annotation as in Fig. 1B. Also, please use the entire space up to the right for the sequence display (I was searching for the FAGSG... start of the CTD you analyzed).*

Response 4: The suggested modification has been made.

Comment 5: *Discussion: It is striking that the heptads in Dm CTD containing an Asn7 are most highly enriched in cis-proline conformations and thus a target for Ssu72. Human CTD has five repeats that contain N7, four of them are clustered between repeats 20 and 30. One could discuss this observation.*

Response 5: The reviewers in general encouraged us to rewrite the discussion of our manuscript to provide both broader and deeper context for the significance of our results. This specific comment has been addressed in the rewritten discussion section, to which we refer the reviewer.

Comment 6: *Likewise, although several groups now reported that Cdk9 predominantly phosphorylate Ser5 of the CTD (also in yeast!), it might be worth mentioning that even under saturated phosphorylation levels (Fig. 2B) no Ser2 phosphorylation occurs. This also adds to the specificity of the kinase that does not phosphorylate any given Ser-Pro motif, meaning Ser2 and Ser5.*

Response 6: The reviewer is also correct to point out this feature of our data, which is now emphasized in the second paragraph of the re-written discussion.

Comment 7: Fig. 2: I don't really understand the bar diagram in panel F. Gray bars indicate the pSer5 or pThr5 position. But why is the blue line that indicates the cis-proline enriched state then always two bars to the right from the gray bar? I expected the blue cis-Pro6 bar next to it.

Response 7: The reviewer has interpreted the results correctly. This plot displays peak movement in the 2D HSQC and CON spectra (Fig. 3) and it is true that the residues in the 5- and 7-positions of the heptad display larger backbone chemical shift perturbations than the Pro6 residues do. Proline is known to show dramatic chemical shift changes in the cis-proline state, but these chemical shift changes are observed for the C β and C γ resonances (in response to re-puckering of the five-member ring). These chemical shift changes are seen clearly in Fig. 5 and constitute the major piece of evidence used to support the assignment of cis-proline conformations for the Ser5-Pro6 peptide planes (and to a lesser extent in the Ser2-Pro3 planes).

Comment 8: Fig. 8D: In this model of an YSPTSPN repeat shouldn't the Ndelta2 nitrogen be the donor for the hydrogen bond with the backbone CO of Thr5 instead of the Odelta1?

Response 8: The orientation of the original figure made it appear that those two atoms are close by, but in the tertiary structure they are actually rather well separated. In order to clarify this point, we remade Figure 8D with an alternative orientation to show that the intramolecular hydrogen bond network. The isomeric state of asparagine is the most energetically favorable one. In this isomeric state, the side chain can adopt the conformation shown in Figure 8D, or it can be repositioned through a 180-degree flip in which the side chain N-delta nitrogen would form the carboxyl group of Thr4. Because the NH-O hydrogen bond is weaker than OH-O hydrogen bond, we expect that the one shown in Figure 8D will be favored.

Comment 9: Lane 91: for clarity you could say: by human P-TEFb.

Response 9: The referenced line now reads "...amino acids in the CTD by human P-TEFb has..." as suggested.

Comment 10: Lane 129: The authors could mention here the median length of the pSer5-pSer5 distance, which seems to be about 18 Ang.

Response 10: The referenced line now reads "... the median pSer5-pSer5 distance (approximately 18Å) is likely to be greater..." as suggested.

Comment 11: Lane 156: Please explain the 15N ZZ-exchange to the general reader.

Response 11: The referenced line now reads "we collected 15N ZZ-exchange NMR spectra, which permits the measurement of conformational exchange on the ms-s timescale, in the presence of *Drosophila* prolyl isomerase Dodo" in response to this important comment.

Reviewer 2

Reviewer 2 has raised several specific points regarding the NMR data presentation, which we address individually below. We especially note that Comment 8 was extremely important and central to the exciting new developments that we have included in the revised manuscript.

Comment 1: *The authors apply NMR and SAXS approaches and some limited sophisticated analyses, however they do not attempt to obtain structural models from their data. Instead, the authors write: “we analyzed the conformation of phosphoryl CTD peptides upon Ssu72 binding.” They use this Ssu-72 bound state to provide a concrete structural model rather than their own experimental data on the free non-phospho and phospho states. They also do not state how this structural analysis was performed; it should be indicated in the text.*

Response 1: Two broad points are addressed in this comment, which we will respond to individually. First, we will address our reasons for not refining a structural model of Dm CTD, either in the unphosphorylated or hyper-Ser5P state. We are confident the reviewer is aware of the technical and physical-chemical challenges inherent to embarking on an atomistic structure determination effort for a system like the CTD, but we will summarize them here for the benefit of others who may be reading this response. Structure determination as normally conceived for cooperatively folded systems is not an option for intrinsically disordered proteins. The reviewer is correct that others have made heroic efforts to assemble enough NMR and SAXS constraints to model ensembles of disordered protein structures, such as the cases of p53 transactivation domain (Blackledge group), and Sic1 (Forman-Kay group). There are several good reasons not to apply a similar strategy to our CTD data. The Flexible Meccano approach utilized in conjunction with the EOM package, or in the ASTEROIDS procedure that relies on software not made publically available by the developers, all build conformers from pools of residue conformations that do not include cis-proline. While the TRaDES engine that generates input structures for the Forman-Kay approach does attempt to provide cis-proline monomers, it is not clear from our experience that it is trained to respond to serine phosphorylation in a way that is appropriate to model the CTD. Alternative approaches, including the very successful MC modelling approach of the Pappu laboratory, are currently unable to accommodate phosphoserine at all. Given these limitations, we elected to provide only general discussion of the overall structure of CTD, while emphasizing the role of Ser5 phosphorylation in controlling Pro6 isomerization. This narrative was critical for establishing the relative substrate preferences of Dm P-TEFb and for our work with Ssu72.

On the reviewer's second comment, the kinetic data for Ssu72 catalyzed dephosphorylation provided a particularly strong rationale for considering the bound-state conformation of Ser5P-enriched CTD, for which there was prior crystallographic data to build from. We followed the reviewer's advice and rewrote the two paragraphs of the manuscript that describe the model in Figure 8 to provide details of our data analysis. In brief summary, we superimposed all four published complex structures to show that the same configuration is adapted by the backbone of the CTD peptides. We then virtually mutated the residues and analyzed the intramolecular interaction of the mutated residues with other residues of the CTD peptide. We considered all possible rotameric states of the sidechain and in each case the one most favored energetically was exhibited in Figure 8.

Comment 2: *Importantly, the context of these structures should be provided (not just as PDB codes in the figure legend) and the citation should be indicated in the text.*

Response 2: The reviewer is correct to point this out. The necessary changes have been made to the text, the relevant portion of which is reproduced below:

Several structures have been published for *Drosophila* or human Ssu72 bound to CTD peptides of different phosphorylation states including *Drosophila* Ssu72, *Drosophila* Ssu72-Symplekin, and human Ssu72-Symplekin, bound to pSer5 CTD (PDB codes: 3P9Y¹⁶, 4IMJ³¹, and 3O2Q³², respectively), and *Drosophila* Ssu72-Symplekin bound to a CTD peptide with Thr4/Ser5 doubly phosphorylated (PDB code: 4IMI³¹). The superimposition of these structures reveals that all known phosphoryl CTD peptides adopt a tight turn facilitated by *cis*-proline upon binding to Ssu72-Symplekin (Fig. 8A).

Comment 3: *It is not clear how stable the heptad structural element is, though the narrow NH dispersion suggest that they are not stable. This will have a big impact on the model. Clearly, cis-trans proline isomerization occurs relatively slowly, but is still on the order of ms. Are the structural elements populated to a degree that hydrogen bonds or NOEs can be observed by NMR.*

Response 3: We interpret “stable” to mean that the conformation adopted in solution by a given heptad persists in time such that the lifetime of a given conformation exceeds the NMR measurement timescale of ms. Given this definition, the reviewer is correct; the narrow NMR lineshapes and poor amide-proton chemical shift dispersion suggest that the tertiary structure of Dm CTD is not temporally stable, but rather that rapid fluctuations interconvert the chain between energetically similar microstates separated only by low barriers (relative to the available thermal energy). Even so, rapid structural fluctuations are not necessarily incompatible with a degree of local order that would support the development of substantial NOE cross-peak intensity between nearby nuclei. The reviewer is correct to speculate that we have attempted to measure NOEs for Dm CTD. Even at low temperature (4 °C) and ultra-high field (850 MHz), we have never observed substantial NOE buildup consistent with persistent structure in Dm CTD. Personal experience on the part of the corresponding author and conversations at both the ENC and the Intrinsically Disordered Proteins GRC suggest that this observation is common to a large fraction of disordered proteins that have been assessed by the community.

In contrast, uncatalyzed proline *cis-trans* isomerization is known to be very slow on the NMR timescale (seconds or longer). That Dm CTD behaves in this way is supported by (1) the observation of two narrow resonance lines for adjacent phosphoserine residues, one for *cis*-proline and the other *trans*-proline and (2) our ability to catalyze the rate of isomerization through the addition of Dodo into a time regime that enabled measurement of exchange through ZZ-spectroscopy.

Comment 4: *Could this be performed using heptad repeat peptides? If the authors think these structural elements are not stable, this should be indicated in the text, since the reader will be heavily influenced by the structures shown in Fig. 8 in the Ssu72-bound state.*

Response 4: Several reports already appear in the literature where short peptides of 1-3 CTD heptads are analyzed structurally by NMR and CD. In general, the authors of these studies conclude that in aqueous solution, when unbound by other biomolecules, CTD consensus heptads adopt spatially heterogeneous and highly dynamic coil-like structures. This work is entirely consistent with our findings on our longer and more sequence-diverse CTD construct. The reviewer is right to raise this concern, and we were in error to not cite more of the peptide studies than we did in our original submission. Our revised manuscript addresses both of these points at

the end of the section describing the heptad-scale NMR structural data we acquired on the unphosphorylated form of CTD Δ 2' in the following way:

Thus, our NMR and SAXS data are strongly consistent with Dm CTD free in solution adopting a spatially heterogeneous ensemble that is highly dynamic on the \sim ns timescale. This observation is consistent with prior reports that concluded short CTD-derived peptides are predominantly random in solution, lacking long-range order or temporally persistent tertiary structure^{14,15,27,28}. In summary, the unphosphorylated state CTD Δ 2' is highly disordered, temporally dynamic, and contains nearly all *trans*-proline.

Comment 5: *What are the implications of the change in the equilibrium populations of the cis-trans proline isomerization states? Are there different binding domains of various downstream partners for these two states? Could there also be a change in exchange rate, with the non-phosphorylated having some cis but in fast exchange?*

Response 5: This question was one among several excellent ones raised by our reviewers regarding the breadth and depth of our discussion section. As stated previously in this letter, the discussion has been completely rewritten in our revised manuscript. We hope that the new version will address the reviewer's point.

Comment 6: *While the phosphorylation does not have a significant impact on the radius of gyration, other properties of the CTD may be influenced. For example, there are several studies indicate that the CTD is involved in formation of hydrogels or liquid droplets. The authors seem to discount this hypothesis, but fail to address it experimentally.*

Response 6: We are sorry that we appear to have mislead the reviewer. We do not seek to discount or discredit the hypothesis that CTD is involved in the formation of hydrogels or liquid droplets under a variety of conditions. Quite to the contrary, we find the papers the reviewer refers to highly compelling. That said, under no conditions explored in the current manuscript have we experienced liquid-liquid phase separation or hydrogel formation for our Dm CTD constructs, but this is entirely consistent with the published literature on the subject. First, we note that unlike the present work, the studies referred to by the reviewer were conducted with CTD constructs that contained the cluster enriched in Lys7-containing heptads, which are likely to be involved in the mechanism of aqueous phase separation. Second, we note that phase separation often occurs in the presence of protein and/or nucleic acid binding partners that were not the subjects of the present study and therefore were absent from our experiments. The reviewer is correct to think that future exploration of this topic in follow-up manuscripts using *Drosophila* CTD constructs has merit.

Comment 7: *Within the context of the full CTD, does elimination of specific phosphorylation sites by mutation have the same effect as elimination of the 1659-1737 region. Does this match the expectations of the model.*

Response 7: We are pleased to see that the reviewer is thinking along the same lines as we are regarding future directions. The laborious work of generating the transgenic fly lines required to thoroughly explore this topic is underway.

Comment 8: *The authors didn't push the phosphorylation reaction to obtain a more homogeneous/uniform sample; they obtain a mix, with a maximum of 10 sites. Newer published*

phosphorylation approaches such as those using ATP regeneration systems or dialysis methods could potentially yield almost uniform 12-site phosphorylation.

Response 8: We acknowledge that our in vitro kinase assays yield a mixture of phosphoisoforms, which is consistent with other reports throughout the literature:

1. E. W. Martin, A. S. Holehouse, C. R. Grace, A. Hughes, R. V. Pappu, T. Mittag, Sequence determinants of the conformational properties of an intrinsically disordered protein prior to and upon multisite phosphorylation, *J. Am. Chem. Soc.* (2016).
2. F. Cordier, A. Chaffotte, E. Terrien, C. Prhaud, F.-X. Theillet, M. Delepierre, M. Lafon, H. Buc, N. Wolff, *J. Am. Chem. Soc.* 2012, 134, 20533–20543
3. Xiang, S., Gapsys, V., Kim, H.-Y., Bessonov, S., Hsiao, H.-H., Möhlmann, S., Klaukien, V., Ficner, R., Becker, S., Urlaub, H., Lührmann, R., de Groot, B., Zweckstetter, M., 2013. Phosphorylation Drives a Dynamic Switch in Serine/Arginine-Rich Proteins. *Structure* 21, 2162–2174. doi:10.1016/j.str.2013.09.014
4. Mittag, T., Marsh, J., Grishaev, A., Orlicky, S., Lin, H., Sicheri, F., Tyers, M., Forman-Kay, J.D., 2010. Structure/function implications in a dynamic complex of the intrinsically disordered Sic1 with the Cdc4 subunit of an SCF ubiquitin ligase. *Structure* 18, 494–506. doi:10.1016/j.str.2010.01.020

However, we took the reviewer's advice and performed the kinase reaction following the dialysis method presented by Bah et. al. (reference 25 of the revised manuscript). The results of this study are summarized here:

Figure for Review: Dm P-TEFb Kinase Reaction Performed Using a Dialysis Method. Each reaction contained 100 μ M CTD Δ 2' with 100 μ g/ml Dm P-TEFb in kinase buffer (50 mM Tris HCl pH 7.5, 50 mM NaCl, 20 mM MgCl₂, 10 mM ATP, and 2 mM DTT) in a volume of 500 μ l. Following enzyme addition samples were loaded into dialysis bags (Spectra/Por 1000 MWCO) and placed in a beaker containing 100 ml of kinase buffer. Samples were suspended in buffer with mild stirring and left at room temperature for ~30 hours. Samples were then removed from dialysis bags purified by C18 spin columns (Pierce) and by ion-exchange (Dowex 50wx8, Sigma) prior to analysis by MALDI-TOF-MS. MS analysis was performed as described in online methods. The results were identical to those reported in the main text.

As the figure clearly shows, the results were identical to those presented in Figure 2B. In addition, we performed extensive kinetic measurements of the Dm P-TEFb kinase reaction using RT-NMR (as described above), which allowed us to determine the apparent rate constants for each phosphorylation event. This revealed that while there is a mixture of phosphoisoforms present at saturation, the sites that we extensively characterized in terms of the proline *cis-trans* equilibria were phosphorylated to the same very high extent (>90%). What is more, we contend that the variation seen is likely to reflect the real sequence-directed biases in phosphate installation by Dm P-TEFb. In short, we are confident that the data presented yield a realistic picture of the fullest extent to which P-TEFb can phosphorylate this region of CTD.

Comment 9: *The finding that Ssu72 dephosphorylates the cis-stabilized sites is not novel since previous structural studies have already identified that aspect of the phosphatase.*

Response 9: The reviewer is correct. Our intent was to demonstrate how the sequence specific switch induced by phosphorylation could impact the activity of a protein.

Comment 10: *The paper lacks excitement, not highlighting how the work is important to a deeper understanding of the function of the protein or the biological roles it plays. The discussion is only a single paragraph long, with little room to make connections between the findings in the paper and important biology.*

Response 10: As summarized above in response to other comments on the discussion, this section has been entirely rewritten and expanded. We hope that the revisions we have made will satisfy the reviewer on this comment as well.

Comment 11: *A minor comment is that “apo” generally applies to a free state lacking a binding partner (protein, nucleic acid, small molecule ligand). As there are no binding partners studies, the authors’ use of this term for the non-phosphorylated state is confusing.*

Response 11: This is a very good point and we have systematically replaced “apo” with “unphosphorylated” in the revised manuscript.

Reviewer 3

We thank Reviewer 3 for the very kind remarks made about our manuscript. Specifically, we note that the reviewer found our results “nicely show that serine phosphorylation affects the cis-trans state of the CTD prolines and that this is affected by the surrounding sequence” and further that the reviewer states “This manuscript represents one of the very few successes in gaining some understanding on the structure of the CTD and on the impact of phosphorylation.” As the reviewer made no specific requests for revision, there are no action points for us to address here.

Reviewers' Comments:

Reviewer #1 (Remarks to the Author):

The revised manuscript by Gibbs et al. is a thoroughly performed revision of the study that addresses almost all points raised from the initial review process. It is a pity that the antibody experiments did not work out, but I can easily understand that the epitopes did not recognize the phosphorylation sites of the Dm CTD properly. The phosphorylation kinetics shown in Fig 2E are very nice and I appreciate the interpretation that the phosphorylation rates fall basically into three classes according to the sequence motifs provided, suggesting a distributive mechanism for the reaction under in vitro conditions. I suggest publication of the manuscript in the present form.

Reviewer #2 (Remarks to the Author):

The authors have substantially responded to the many of our comments on the paper. The new discussion is a major improvement over the previous discussion. The authors have not considered the structural ensemble of the CTD, but this is outside the scope of the paper. The resulting manuscript provides a valuable contribution to the field.

We note the following concern, which should not prohibit publication but should be discussed.

“Based on these results, we propose a model in which hyper-pSer5 does little to alter the global-scale structure of the CTD Δ 2' region of CTD. Instead, dramatic structural rearrangements occur on the single heptad scale, driven by sequence context-dependent proline trans-to-cis isomerizations (Fig. 6B).”

It should be noted that the only data that the authors have with regards to the “global-scale structure” is the unaffected radius of gyration obtained from the SAX data. So the authors should consider rephrasing this to “to alter the radius of gyration of the CTD Δ 2'”. As we suggested in our initial review, the work of others suggests that bulk properties of the CTD are affected by phosphorylation. Furthermore these changes in bulk CTD properties are suspected to be linked to function.

Other comments:

- The text would benefit from some polishing.
- There is a problem with the nomenclature “CTD Δ 2' “ and “CTD Δ 2”. In figure 1, “CTD Δ 2” refers both to the large construct with 34 repeats but missing the conserved 56 amino acids (fig 1A) and to the smaller fragment that contains these conserved residues (fig 1E). The “CTD Δ 2' “ then is a longer version of the second type (1657-1739, fig 1E). This is

confusing and needs fixing. Related to this, Figure 1B should have amino acid numbers for the deletions listed in the figure legend.

- Figures 2F is still confusing.

Reviewer 1

Reviewer 1 suggested publication of the manuscript in its present form, so there are no specific requests to address. We share the reviewer's disappointment that the antibody study previously recommended did not pan out, but are glad that our efforts were deemed satisfactory. We thank Reviewer 1 for working with us on this manuscript.

Reviewer 2

We are also grateful that Reviewer 2 both provided us with excellent suggestions for revision and was satisfied with our efforts to respond. Reviewer 2 did have a few lingering questions about the revised manuscript, which we address individually below.

Comment 1: *“Based on these results, we propose a model in which hyper-pSer5 does little to alter the global-scale structure of the CTD Δ 2' region of CTD. Instead, dramatic structural rearrangements occur on the single heptad scale, driven by sequence context-dependent proline trans-to-cis isomerizations (Fig. 6B).”*

It should be noted that the only data that the authors have with regards to the “global-scale structure” is the unaffected radius of gyration obtained from the SAX data. So the authors should consider rephrasing this to “to alter the radius of gyration of the CTD Δ 2'”. As we suggested in our initial review, the work of others suggests that bulk properties of the CTD are affected by phosphorylation. Furthermore these changes in bulk CTD properties are suspected to be linked to function.

Response 1: The main text has been changed to read “Based on these results, we propose a model in which hyper-pSer5 does little to alter the scaling properties of the CTD2' region of the *Drosophila* CTD; specifically, the measured R_g and D_{max} are unaltered.

Comment 2: The text would benefit from some polishing.

Response 2: We hope that the final revisions and editorial process will address this concern.

Comment 3: *There is a problem with the nomenclature “CTDdelta2'” and “CTDdelta2”. In figure 1, “CTDdelta2” refers both to the large construct with 34 repeats but missing the conserved 56 amino acids (fig 1A) and to the smaller fragment that contains these conserved residues (fig 1E). The “CTDdelta2'” then is a longer version of the second type (1657-1739, fig 1E). This is confusing and needs fixing. Related to this, Figure 1B should have amino acid numbers for the deletions listed in the figure legend.*

Response 3: The amino acid numbers for each mutant have been added to Fig 1b. In Fig 1e, we have attempted to remove the confusion while using nomenclature that draws a strong connection between the region removed in CTD Δ 2 fly line and the recombinant protein construct. The (hypothetical) recombinant construct that codes for the deleted residues is now referred to as CTD2, while the expanded and biophysically tractable construct is named CTD2' throughout the manuscript. We hope that removing the Delta from the recombinant construct's name will avoid confusion.